# Giant nonvolatile manipulation of magnetoresistance in magnetic tunnel junctions by electric fields via magnetoelectric coupling

Aitian Chen [1,2,3], Yan Wen[3], Bin Fang[4], Yuelei Zhao[3], Qiang Zhang[3], Yuansi Chang[5], Peisen Li[1,6], Hao Wu [5], Haoliang Huang [7], Yalin Lu [7], Zhongming Zeng[4], Jianwang Cai[5], Xiufeng Han[5], Tom Wu[3], Xi-Xiang Zhang [3] & Yonggang Zhao [1,2]

Electrically switchable magnetization is considered a milestone in the development of ultralow power spintronic devices, and it has been a long sought-after goal for electric-field control of magnetoresistance in magnetic tunnel junctions with ultralow power consumption. Here, through integrating spintronics and multiferroics, we investigate MgO-based magnetic tunnel junctions on ferroelectric substrate with a high tunnel magnetoresistance ratio of 235%. A giant, reversible and nonvolatile electric-field manipulation of magnetoresistance to about 55% is realized at room temperature without the assistance of a magnetic field. Through strain-mediated magnetoelectric coupling, the electric field modifies the magnetic anisotropy of the free layer leading to its magnetization rotation so that the relative magnetization configuration of the magnetic tunnel junction can be efficiently modulated. Our findings offer significant fundamental insight into information storage using electric writing and magnetic reading and represent a crucial step towards low-power spintronic devices.

[1] Department of Physics and State Key Laboratory of Low-Dimensional Quantum Physics, Tsinghua University, Beijing 100084, China. [2] Collaborative Innovation Center of Quantum Matter, Beijing 100084, China. [3] Physical Science and Engineering Division, King Abdullah University of Science and Technology, Thuwal 23955-6900, Saudi Arabia. [4] Key Laboratory of Nanodevices and Applications, Suzhou Institute of Nano-tech and Nano-bionics, Chinese Academy of Sciences, Ruoshui Road 398, Suzhou 215123, China. [5] Beijing National Laboratory for Condensed Matter Physics, Chinese Academy of Sciences, Beijing 100190, China. [6] College of Mechatronics and Automation, National University of Defense Technology, Changsha 410073, China. [7] Hefei National Laboratory for Physical Sciences at the Microscale & National Synchrotron Radiation Laboratory, University of Science and Technology of China, Hefei 230026, China. These authors contributed equally: Aitian Chen, Yan Wen. Correspondence and requests for materials should be addressed to T.W. (email: tao.wu@kaust.edu.sa) or to X.-X.Z. (email: xixiang.zhang@kaust.edu.sa) or to Y.Z. (email: ygzhao@tsinghua.edu.cn)

Developing high-density, high-speed, and low-power memory is becoming imperative due to the ever-growing demands for data storage and the increasing gap between processor and off-chip memory speeds[1,2]. Magnetoresistive random access memory (MRAM), whose key attributes are non-volatility, infinite endurance and fast random access[3,4], is currently the most promising contender for next-generation memory and universal memory[4–7]. A magnetic tunnel junction (MTJ) with large tunnel magnetoresistance (TMR)[8,9], which has two ferromagnetic (FM) layers sandwiching a thin insulating barrier, has been chosen as the MRAM storage element because of its large signal for the read operation[3,7]. In pursuit of lower power consumption, a tremendous flurry of research activities has focused on controlling MTJs with electric fields[10–14] instead of traditional magnetic fields or spin transfer torque effect with high electric current density[15]. Utilizing electric-field-induced change of coercivity or magnetization precession, both Wang et al.[12] and Shiota et al.[13] realized voltage control of magnetoresistance (MR) in MTJs. However, in their work, a bias magnetic field is indispensable to ensure the asymmetric potential wells or fix the precessional axis[16] and, more importantly, the high voltage (nearly 1 V) directly applied on the junctions is close to the breakdown voltage[4,17].

Recently, multiferroic materials[18–21], whose technological appeal is the ability to control magnetism with electric fields, have revived the magnetoelectric research[22]. Integrating spintronics and multiferroics is emerging as a leading candidate for memory and logic[23], and opens a new avenue for exploring energy-efficient electric-field-controlled MTJs. Efforts have been made toward employing an ultrathin ferroelectric (FE) or multiferroic layer as the tunnel barrier to modify the MR of MTJs based on FE-polarization-reversal-induced variation of interfacial spin polarization[24–26]. However, an ultrathin barrier layer with only a few nanometers thick is very difficult to keep ferroelectric[14,27]. Moreover, operation at cryogenic temperature is detrimental for practical applications. To overcome these drawbacks, a scheme of MTJ stacks grown on FE substrate was proposed theoretically[11,28,29], which relied on the recent progress in the room temperature electric-field control of magnetism in FM/FE multiferroic heterostructures[21,30–40]. Through strain-mediated magnetoelectric coupling, MR of MTJs can be electrically controlled by rotating magnetization of the free layer. Based on this scheme, an ultrahigh storage capacity, ultralow power dissipation and high-speed MRAM device has been proposed[11]. The three-terminal design of this scheme separates the write and read cells[41], and the voltage is not exerted on the MTJ to avoid damaging it. However, this geometry complicates the fabrication process[14] in comparison with the more common two-terminal MTJs. To date, only a few experimental trials with small and volatile manipulation of MR in MTJs by electric fields have been reported[42,43]. Non-volatile electrically controlled magnetism[44–47] was demonstrated in multiferroic heterostructures, however, it has not been used in spintronic devices. Therefore, room temperature, giant, and non-volatile electrical manipulation of MR in MTJs without a bias magnetic field is highly desirable, while still elusive, though it would be a crucial step towards practical devices such as MRAM.

Here, we report room temperature, giant, and non-volatile manipulation of MR in the MgO-based MTJs solely by electric fields without the assistance of a magnetic field using MTJs/FE configuration. Through integrating spintronics and multiferroics, high quality MTJs based on MgO tunnel barriers were deposited on $Pb(Mg_{1/3}Nb_{2/3})_{0.7}Ti_{0.3}O_3$ (PMN-PT) and a giant non-volatile electric-field-controlled MR in MTJs with a relative change of about 55% was achieved at zero magnetic field, which can be ascribed to the electric-field-induced magnetization rotation of the free layer via strain-mediated magnetoelectric coupling. This non-volatile electric-field manipulation of MR in MTJs is significant for developing future spintronic devices with ultralow energy consumption and realizing information storage with electric writing and magnetic reading.

## Results

**Non-volatile electrical manipulation of MR in MTJs.** The detailed structure, composition, and configuration of the devices are schematically shown in Fig. 1a, in which the multilayers of the MTJ stack, Ta(5 nm)/Ru(5 nm)/IrMn(8 nm)/CoFe(2.3 nm)/Ru(0.85 nm)/CoFeB(2.6 nm)/MgO(2.3 nm)/CoFeB(2.6 nm)/Ta(5 nm)/Ru(7 nm), was deposited on PMN-PT using Singulus sputtering at room temperature. We imaged the cross-sections of the devices used in this work (same batch) using high-resolution transmission electron microscopy (HRTEM) (Supplementary Fig. 1), which confirmed experimentally that the MTJs had the designed structure. The PMN-PT with (011) orientation was chosen as the active element in the device, because of its large piezoelectric coefficients[48]. To enhance the performance of the devices, we used both the antiferromagnetic layer IrMn and the artificial antiferromagnetic structure of CoFe/Ru/CoFeB. This design is different from the studies previously reported in which no antiferromagnetic layer was employed[43]. With this particular design, at zero magnetic field, the top CoFeB layer acts as a free layer whose magnetization can be tuned by electric fields, whereas the magnetization of the bottom CoFeB layer is fixed due to the pinning effect of the antiferromagnetic IrMn layer. The magnetization was pinned along the [100] direction of the PMN-PT (011) single crystal (Fig. 1b) by a constant magnetic field applied during annealing. As shown in Fig. 1a, the electric field was applied only to the FE substrate to produce anisotropic strain through to the bottom Au electrode, and was not applied to the MTJ element to avoid the potential damage to the MTJs. The circular-disk-shaped MTJ devices of 10 μm in diameter were fabricated using photolithography and the resistance-area product (RA) at low resistive state is 23 Ω μm². The TMR ratio was as high as about 235% for our devices, which is comparable to that of MTJs grown on silicon wafers (Supplementary Fig. 2) and is much higher than those obtained from MTJs deposited on the FE substrates[42,43], suggesting the high quality of our MTJ devices.

To explore the electric effects on the MTJs, the MR curves were measured at room temperature with different electric fields and a striking effect of the electric field on MR was observed (Supplementary Fig. 3a). As the results presented in Fig. 1c illustrate, the behaviors of the magnetic field dependence of RA are completely different at $E = 0\,kV\,cm^{-1}$ by following application of electric fields of $8\,kV\,cm^{-1}$ (referred as $+0\,kV\,cm^{-1}$) and $-1.6\,kV\,cm^{-1}$ (referred as $-0\,kV\,cm^{-1}$) respectively, suggesting that the electric modulation is giant and non-volatile. The reason to select $8\,kV\,cm^{-1}$ and $-1.6\,kV\,cm^{-1}$ will be mentioned later in the study of ferroelectric property and the strain property of the PMN-PT (011) single crystal. Around $H = 0$ Oe, there were two high and low resistance states for $E = +0\,kV\,cm^{-1}$, which correspond to the nearly antiparallel and parallel magnetization configurations of the two CoFeB layers (inset of Fig. 1c), respectively. Since the RA value of $E = -0\,kV\,cm^{-1}$ around $H = 0$ Oe is intermediate between the two high and low resistance states, the alignment of the magnetic moments of the two CoFeB layers is non-collinear, suggesting magnetization rotation of the free layer (Supplementary Fig. 4). Thus, the magnetic configuration is reversibly switched between antiparallel and non-collinear (case I) and between parallel and non-collinear (case II) as illustrated by the blue and pink double-headed arrows in Fig. 1c, respectively. According to Julliere's model[28,49], the values of rotation angle $\varphi$ of the free layer can be obtained from the

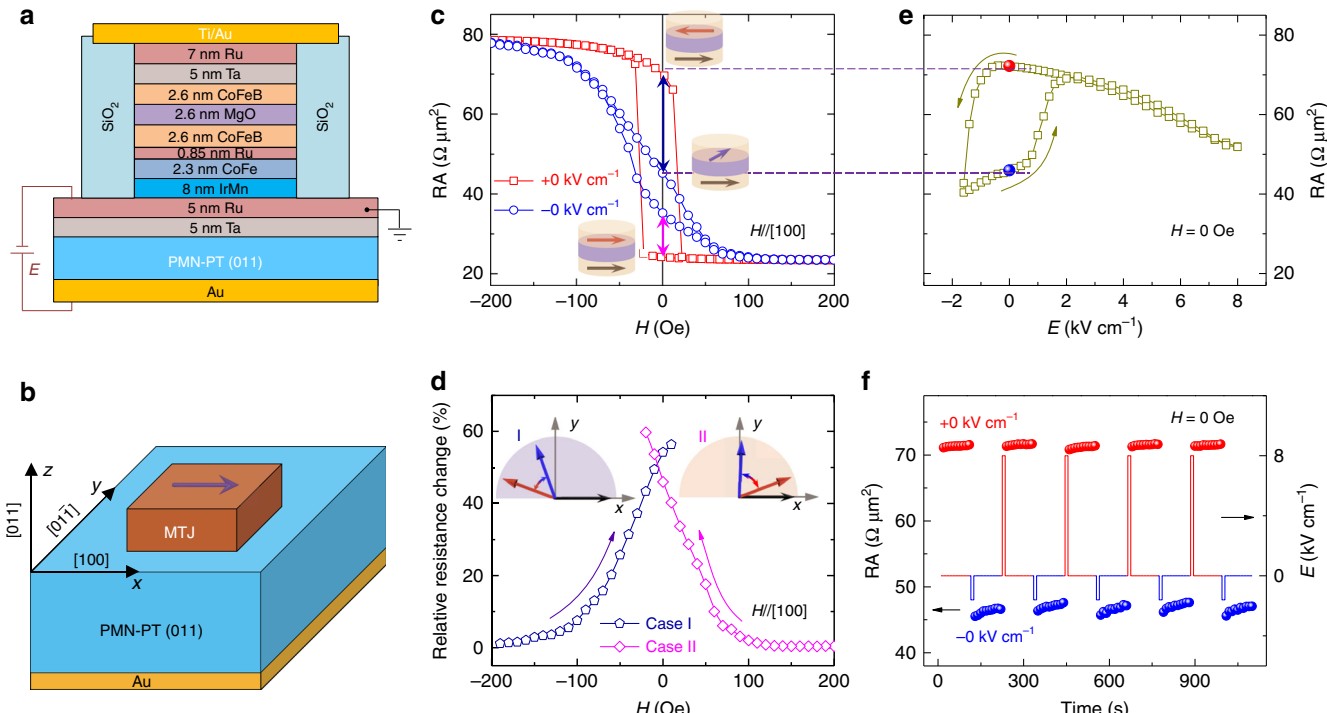

**Fig. 1** Non-volatile electrical manipulation of MTJs. **a** Schematic of the MTJ device structure deposited on PMN-PT. Voltage was exerted on the FE substrate to avoid damaging the MTJs. **b** Crystal orientation of the PMN-PT (011). The pinning direction of MTJ is along the [100] direction, represented by the purple arrow. $x$, $y$, and $z$ edges denote the pseudocubic [100], [01$\bar{1}$], and [011] crystallographic directions of PMN-PT, respectively. **c** MR curves measured at $E = \pm 0$ kV cm$^{-1}$ after applying 8 kV cm$^{-1}$ and $-1.6$ kV cm$^{-1}$, respectively. The RA plotted here is the tunnel resistance for a circular shape with a 10 μm diameter. The insets illustrate the relative magnetization alignment around zero magnetic field. **d** Relative resistance change between $E = \pm 0$ kV cm$^{-1}$ for case I and case II, illustrated by purple and red double-headed arrows in **c** respectively. The insets show schematic of magnetization configuration for case I and case II. The dark arrow denotes the magnetization of the pinned layer, which is fixed around $H = 0$ Oe. The red and blue arrows denote the magnetization of the free layer at $E = \pm 0$ kV cm$^{-1}$, respectively. **e** Dependence of RA on electric field under $H = 0$ Oe for case I. The two distinctive resistive states at $E = 0$ kV cm$^{-1}$ indicate non-volatile and reversible electrical manipulation of MR in MTJs. **f** Repeatable bistable resistance states modulated by 8 kV cm$^{-1}$ and $-1.6$ kV cm$^{-1}$ electric-field pulses in the absence of a bias magnetic field

corresponding resistance (Supplementary Note 1 and Supplementary Table 1). It is notable that magnetization configurations for $E = +0$ kV cm$^{-1}$ at $H = 0$ Oe are nearly antiparallel and parallel with a small angle. For simplicity, the nearly antiparallel/parallel is denoted as antiparallel/parallel. Figure 1d presents the relative resistance change (Methods) at $E = \pm 0$ kV cm$^{-1}$ for case I and case II, respectively, along with their corresponding magnetization configuration. It is evident that MR can be tuned by electric fields under a bias magnetic field ranging from about −100 to 100 Oe. Most importantly, the manipulation of MR with a 55% change was realized by electric fields even at zero magnetic field. We took Case I as an example to further explore how to manipulate MR of MTJs using electric fields in our devices and the electric-field dependence of RA was measured at zero magnetic field with closed loop-like behavior as shown in Fig. 1e. Before applying electric fields, the device was initialized to the high resistance state with an antiparallel magnetization configuration after applying a magnetic field of −300 Oe. When sweeping the electric field from 8 kV cm$^{-1}$ to −1.6 kV cm$^{-1}$, the resistance changed dramatically at around ±1.6 kV cm$^{-1}$, which is close to the coercive electric field ($E_C$) of PMN-PT, implying its close relationship with the FE polarization switching as discussed later. As expected, at $E = 0$ kV cm$^{-1}$, two distinctive resistance states with a large difference indicate clearly that the giant, non-volatile and reversible electrical manipulation of MR in the MTJ has been achieved. To re-confirm the reversible and non-volatile switching of the two resistance states, we performed the switching experiments using 8 kV cm$^{-1}$ and −1.6 kV cm$^{-1}$ electric-field

pulses as shown in Fig. 1f. It is evident that a reversible, non-volatile and giant modulation of MR in the MTJ was achieved at room temperature without the assistance of a magnetic field. A slow relaxation process is observed after applying electric-field pulses which was also reported in the previous literatures[42,43] and may originate from the release process of the charges induced by ferroelectric polarization.

Note that this giant and non-volatile electrical manipulation of MR in MTJs is sharply distinguished from the volatile modulation observed for the MTJs/FE system[42,43] previously. Moreover, our devices work stably at room temperature and are much more suitable for practical applications than the MTJs that have a tunnel barrier of an ultrathin FE or multiferroic layer, which require a much lower working temperature than room temperature[24–26]. Our results are also different from the electric-field-assisted switching in MTJs[12,13], where a bias magnetic field is indispensable.

**Non-volatile electric-field control of magnetism in MTJs.** As the resistance of MTJ is closely related to the relative magnetization orientation of the free layer and pinned layer in MTJ, a careful magnetic characterization of MTJ multilayers was performed with in situ electric fields (Supplementary Fig. 5). Figure 2a shows the magnetic hysteresis (M-H) loops measured with a magnetic field applied along the [100] direction at $E = \pm 0$ kV cm$^{-1}$. The M-H curve obtained at $E = +0$ kV cm$^{-1}$ shows a typical M-H loop for an MTJ stack in which the magnetizations

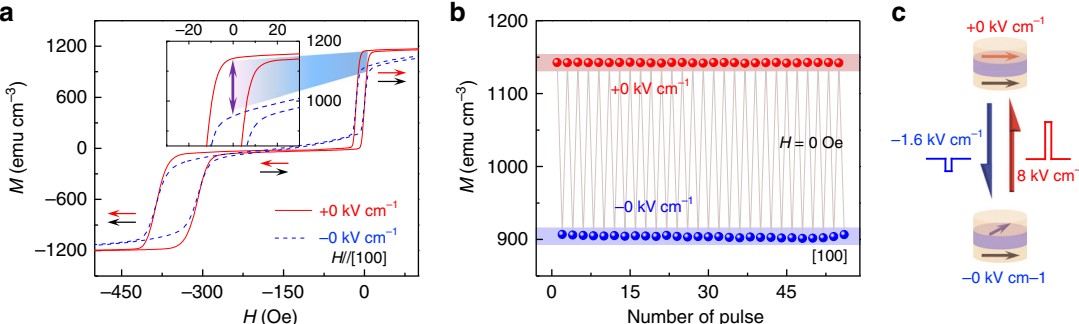

**Fig. 2** M-H loops of MTJ under electric fields with applied magnetic field along the [100] direction. **a** M-H loops of MTJ for $E = \pm 0$ kV cm$^{-1}$, respectively. The arrows denote the magnetization configurations for corresponding magnetic fields. The inset presents electric-field-induced magnetization variation of the free layer around zero magnetic field. **b** Reversible and remarkable magnetization switching induced by successive application of electric-field pulses at $H = 0$ Oe for case II in Fig. 1d. **c** The magnetization configurations of MTJ at $E = \pm 0$ kV cm$^{-1}$ in **b**. These two magnetization states can be reversibly switched by 8 kV cm$^{-1}$ and –1.6 kV cm$^{-1}$ electric-field pulses

in both the CoFeB layers align in the film plane, resulting in full saturation of magnetization at the parallel state and nearly field-independent zero magnetization at the antiparallel state (three flat parts in the curve). However, the hysteresis loop measured at $E = -0$ kV cm$^{-1}$ shows a different behavior in the same magnetic field range. The smaller squareness of the M-H loop and the smoother magnetization change at $E = -0$ kV cm$^{-1}$ than those at $E = +0$ kV cm$^{-1}$ are ascribed to the electrical modulation of magnetic anisotropy in both the FM layers. Around zero magnetic field, the magnetization of the pinned layer is fixed in the [100] direction by the exchange bias, which is not affected by electric fields[50], while only the magnetization of the free layer can be rotated by electric fields leading to the non-collinear magnetization configuration (Fig. 1d). To demonstrate clearly the magnetization rotation by electric fields, we took Case II (Fig. 1d) as an example, whose behavior is similar to case I. The variation of magnetization under a series of –1.6/+8 kV cm$^{-1}$ electric-field pulses at $H = 0$ Oe was measured after setting the sample to the parallel state by applying a magnetic field of 1000 Oe (Fig. 2b). As illustrated in Fig. 2c, an electric-field pulse of –1.6 kV cm$^{-1}$ rotates the magnetization of the free layer resulting in a non-collinear magnetization configuration. Then, an electric-field pulse of 8 kV cm$^{-1}$ can switch the magnetization back to the initial parallel state. It is evident from the results illustrated in Fig. 2b that no resetting was needed during the application of the electric-field pulses. More importantly, the magnetization values at $E = \pm 0$ kV cm$^{-1}$ remained almost unchanged over the whole process, which confirms again that the electric-field modulation is non-volatile and reversible. Hence, the remarkable parallel/non-collinear magnetization configurations could be toggled by a set of asymmetric positive/negative intermittent electric fields, which can reversibly control MR in MTJs.

**Non-volatile strain property of PMN-PT (011).** The giant non-volatile electric-field manipulation of MR in MTJs is the direct consequence of the electric-field-tuned magnetic anisotropy of the free layer in MTJs. In FM/FE multiferroic heterostructures, both strain- and charge-mediated magnetoelectric couplings are expected. The charge effect, which affects only several nanometers, can be ruled out due to the metal multilayers inserted between PMN-PT and CoFeB[51]. To gain a deeper understanding of the magnetoelectric coupling in our samples, both the ferroelectric property and the strain property of the PMN-PT (011) were investigated (Supplementary Fig. 6), since the strain is closely related to the FE polarization state[45,46,52]. Figure 3a shows the polarization versus electric field (P-E) hysteresis loops of PMN-PT, which reveals that the $E_C$ was about 2 kV cm$^{-1}$. When

applying symmetric bipolar electric fields within ±8 kV cm$^{-1}$, a P-E loop of PMN-PT presents a typical hysteresis loop. The corresponding strains $\varepsilon_x$ and $\varepsilon_y$ versus electric field curves measured along the [100] and [01$\bar{1}$] crystallographic directions of PMN-PT, respectively, display normal butterfly-like behaviors, with two peaks around $E_C$ and without remanent strain (Supplementary Fig. 6b). As long as the electric field exceeds $E_C$, FE polarization was switched to the out-of-plane. Consequently, the large strains around $E_C$ dramatically descend, leading to the peaks, which indicates the close relationship between the strain peaks and FE polarization switching. Alternatively, when measured between 8 kV cm$^{-1}$ and –1.6 kV cm$^{-1}$ in Fig. 3b, the strain peak can persist only if the electric field almost reaches $E_C$ (but remains smaller than $E_C$) and turns back instead, leading to a loop-like non-volatile strain. Due to the converse piezoelectric effect, an electric field induces tensile and compressive strain along the [01$\bar{1}$] and [100] direction, respectively, in consideration of their opposite piezoelectric coefficients[53]. The effective anisotropic strain $\varepsilon_y - \varepsilon_x$ that is responsible for the modification of magnetic anisotropy of the free layer is shown in Supplementary Fig. 6c. Note that the electric-field-dependent RA in Fig. 1e displays a trend similar to the anisotropic strain curve in Supplementary Fig. 6c, which indicates the important role of strain-mediated magnetoelectric coupling. The polarization under a corresponding electric field range with a minor loop is shown in Fig. 3a. After applying electric field 8 kV cm$^{-1}$, the polarization points to the out-of-plane direction with a large polarization at $E = +0$ kV cm$^{-1}$ (Fig. 3c). The polarization then dramatically changes to small values around $E_C$, revealing the in-plane FE polarization, which remains almost the same while sweeping the electric field from –1.6 kV cm$^{-1}$ to –0 kV cm$^{-1}$ (Fig. 3d). Further increasing the electric field from 0 to 8 kV cm$^{-1}$, the polarization is gradually saturated and goes back to the out-of-plane again at +0 kV cm$^{-1}$. This evolution of FE polarization has also been confirmed by piezoresponse force microscopy (PFM)[54,55]. Thus, this asymmetric electric fields technique (minor loop) is effective at stabilizing the metastable in-plane FE polarization state and achieving bistable polarization states at zero electric field, as illustrated in Fig. 3c, d. This reversible conversion between the out-of-plane (+0 kV cm$^{-1}$) and in-plane (–0 kV cm$^{-1}$) polarization of PMN-PT is the origin of the non-volatile strain in Fig. 3b, which has been used to control only FM layer's magnetism[46], ferromagnetic resonance[54], Verwey transition[55], etc. This non-volatile strain is reversible and stable with a good endurance (Supplementary Figs. 6d, 6e and 6f), suggesting the well reproducible ferroelectric domain switching between the in-plane and the out-of-plane.

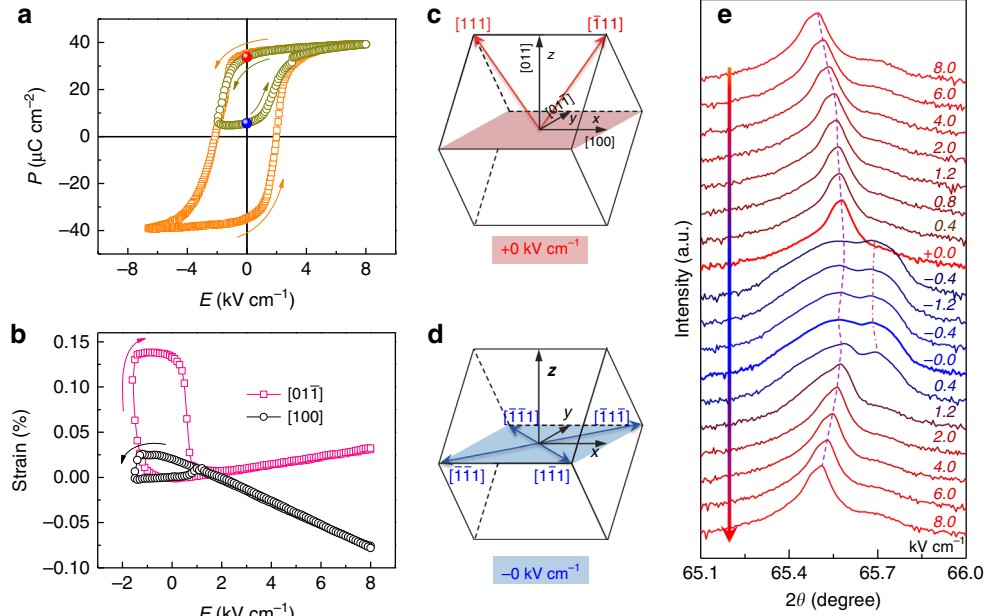

**Fig. 3** Non-volatile strain property of PMN-PT (011). **a** P-E loops of PMN-PT with symmetric and asymmetric bipolar electric fields. **b** Non-volatile piezostrain $\varepsilon_x$ and $\varepsilon_y$ versus electric field curves along the [100] and [01$\bar{1}$] directions, respectively, with asymmetric bipolar electric fields. The arrows in **a** and **b** indicate the electric-field sweeping directions. **c**, **d** Illustration of the FE polarization structure of PMN-PT for $E = \pm 0$ kV cm$^{-1}$, respectively. The rhombohedral crystal structure of PMN-PT (011) has eight spontaneous ferroelectric polarization directions along the <111> directions (the body diagonals of the pseudocubic unit cell); that is, four possible polarization directions lie in the (011) plane, while two point upward and two downward, respectively. **e** Evolution of the PMN-PT (022) diffraction peaks with varying electric fields, in which the electric field swept following the procedure indicated by the arrow

Additionally, the different polarization states at $E = \pm 0$ kV cm$^{-1}$ have different out-of-plane lattice parameters, so that the polarization switching can be identified by X-ray diffraction (XRD) with in situ electric fields. Figure 3e shows the evolution of (022) diffraction peaks of PMN-PT under various electric fields. Electric field sweeping from 8 kV cm$^{-1}$ to $+0$ kV cm$^{-1}$ compresses the out-of-plane lattice and expands the in-plane lattice as revealed by the peak shift to higher angles. Subsequently, the main peak splits into double peaks, suggesting the existence of in-plane polarization[56], until the positive electric field switches the polarization to out-of-plane. Thus, the XRD pattern at $E = 0$ kV cm$^{-1}$ depends strongly on the poling history and has two different states, leading to the non-volatile strain in Fig. 3b. It has been reported that up to 90% of the FE domains can be switched to in-plane state, contributing to this non-volatile strain[54]. Importantly, this non-volatile strain can effectively tune the magnetic anisotropy of CoFeB film (Supplementary Figs. 7 and 8 and Supplementary Note 2), which is certainly the cornerstone of reversible and reliable non-volatile electrical manipulation of MR of MTJs in Fig. 1f.

## Discussion

We have demonstrated a giant (55%), reversible and non-volatile manipulation of MR at room temperature in MTJs/FE system solely by electric fields without the assistance of a magnetic field. This was achieved by the electric-field-induced magnetization rotation of the free layer via strain-mediated magnetoelectric coupling. Our work experimentally realizes non-volatile electrical manipulation of MTJs by integrating multiferroics and spintronics, and represents a crucial step towards electric-field-driven spintronic devices with ultralow power consumption, such as, MRAM and logic elements. Moreover, high quality PMN-PT epitaxial thin films on Si wafers with giant piezoelectricity[57] have been reported, which offers a prospective approach to integrate MTJs/FE structure with current silicon-based electronics, making the magnetic-field-free giant non-volatile electrical manipulation

of MTJs more attractive for future applications with fast electric writing (Supplementary Fig. 9 and Supplementary Note 3) and low energy dissipation (Supplementary Note 4).

## Methods

**Sample fabrication.** The MTJ multilayer films for magnetotransport measurements consisted of, from the substrate side, Ta(5)/Ru(5)/IrMn(8)/CoFe(2.3)/Ru (0.85)CoFeB(2.6)/MgO(2.3)/CoFeB(2.6)/Ta(5)/Ru(7) (numbers are nominal thicknesses in nanometers) and were deposited on PMN-PT (011) single crystals with a size of $10 \times 10 \times 0.5$ mm$^3$. The films were processed into a circular shape with a 10 µm diameter by photolithography and ion milling. Subsequently, the MTJ devices were annealed in a vacuum at 360 °C for 1 h with a magnetic field of 8000 Oe along the [100] crystal axis of PMN-PT. The cross-section of the annealed sample was characterized by HRTEM (Titan 80–300, FEI) equipped with a spherical aberration corrector. In order to simplify the magnetic measurements, the MTJ stack sequence in Fig. 2 is as follows: Ta(5)/Cu(10)/Ta(5)/IrMn(8)/CoFeB(4)/MgO(2.2)/CoFeB(4)/Ta(5)/Cu(5)/Ta(5) (numbers are nominal thicknesses in nanometers). It only has two 4 nm FM layers, so it is easy to measure, and the M-H loops of the pinned layer and free layer are separated. After deposition, the multilayers were annealed in vacuum at 350 °C for 30 min with a magnetic field of 1000 Oe along the [100] crystal axis of PMN-PT. The CoFe, CoFeB, and IrMn denote $Co_{70}Fe_{30}$, $Co_{40}Fe_{40}B_{20}$, and $Ir_{20}Mn_{80}$ alloy with nominal target compositions, respectively. All the stack structures in this study were deposited by a multisource high-vacuum magnetron sputtering system with a base vacuum of $1 \times 10^{-6}$ Pa. An Au layer with a thickness of 300 nm was sputtered on the bottom of PMN-PT as an electrode.

**Relative resistance change.** The relative resistance change in Fig. 1d was deduced from Fig. 1c as follows. For case I, the resistance at $+0$ kV cm$^{-1}$ is larger than that at $-0$ kV cm$^{-1}$ and the relative resistance change is $[R(+0) - R(-0)]/R(-0)$. For case II, the resistance at $+0$ kV cm$^{-1}$ is smaller than that at $-0$ kV cm$^{-1}$ and the relative resistance change is $[R(-0) - R(+0)]/R(+0)$. $R(+0)$ and $R(-0)$ denote the resistance for $E = +0$ kV cm$^{-1}$ and $E = -0$ kV cm$^{-1}$ at a certain magnetic field, respectively. The relative resistance change shows the modulation of magnetoresistance in MTJs by electric fields.

**Magnetic and magnetotransport measurements.** Electric-field control of magnetism was performed using a Quantum Design magnetic property measurement system (MPMS) with in situ electric fields. The resistance of MTJs was acquired via a homemade electromagnet system using the four-probe method with a Keithley 6221 current source and a Keithley 2182 nanovolt meter. The voltage applied to the

PMN-PT was generated by a Keithley 6517 electrometer. The duration of the used voltage pulses in our present work is about 5 s and the width of voltage pulses in theory can be less than 20 ns for our devices (Supplementary Note 3). Furthermore, this duration of the voltage pulses can be less than 10 ns (ref. [11]) using FE films, which needs to be explored in future experiments. All the measurements were carried out at room temperature.

**Characterizations of the PMN-PT**. The P-E loops of PMN-PT were measured using a Radiant Technologies Precision Premier II system. The strain properties were measured using a strain gauge glued on the PMN-PT[58]. XRD with in situ electric fields was characterized by a Rigaku SmartLab 3 kW X-ray diffractometer with a Cu Kα radiation. The morphology was characterized by atomic force microscopy using a commercial scanning probe microscope system (Multimode 8 SPM, Bruker).

## Data availability
The data that support the findings of this study are available from the corresponding author upon reasonable request.

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

## Acknowledgements

The authors thank Dr. Hai Zhong for technical support and Dr. Dalai Li for valuable discussions. This work was supported by the Science Center of the National Science Foundation of China (Grant No. 51788104), the 973 Project of the Ministry of Science and Technology of China (Grant No. 2015CB921402), the National Science Foundation of China (Grant Nos. 51831005, 51572150, 11604384), the National Key Research and Development Program of China (Grant No. 2017YFA0206200), the State Key Laboratory of Low-Dimensional Quantum Physics (Grant Nos. ZZ201701, KF201717) and King Abdullah University of Science and Technology (KAUST).

## Author contributions

A.C. performed the magnetic and magnetotransport measurements under the guidance of Yonggang Z. and T.W. Y.W., B.F., Yuelei Z., and Y.C. deposited MTJ multilayers and fabricated the devices under the guidance of X.Z., Z.Z., and J.C. Q.Z. performed the HRTEM measurements. P.L. performed the strain and P-E loops measurements, Landau-Lifshitz-Gilbert (LLG) equation and micromagnetic simulations. H.W. and X.H. supplied the CoFeB films. H.H. conducted the XRD measurements under the guidance of Y.L. Yonggang Z. conceived and supervised the research. A.C., X.Z., and Yonggang Z. analyzed the data and wrote the manuscript. All authors read and commented on the manuscript.

## Additional information

**Competing interests:** The authors declare no competing interests.

