## [Peer Review File · Nature Communications]

Reviewers' comments:

Reviewer #1 (Remarks to the Author):

The authors study the strain induced, reversible, nonvolatile electric field control of magnetization in a magnetic tunnel junction on a ferroelectric substrate. Most importantly they show a significant change in the magnetoresistance at zero field solely by application of voltage. I think these are interesting and significant experimental findings in the very technologically relevant field of low power voltage controlled spin memory devices. Therefore I recommend publication in Nature Communications after some necessary revisions are made. The issues and comments are the following:

1. One problem with ferroelectric devices that the authors do not discuss is the fatigue resulting from strain mediated operations, which leads to device failure after some operations. Has the durability of these MTJ on ferroelectric devices been tested experimentally? In Fig 2b, ~50 cycles are shown, which is not enough. The durability issue is a weakness compared to "conventional" MTJ devices (e.g. Nat. Mater. 11, 64; Nat. Mater. 11, 39) which do not involve strain induced lattice distortions.
2. What is the speed of the electric field manipulation of magnetoresistance in this device? In other words, how long a voltage pulse is required for this switching to occur? This is important in determining if these devices can really be utilized for low power switching, since the energy required in MTJ operations is current \times voltage \times time. The timescale is Fig. 1f (~seconds) suggests that this operation might be too slow for practical low power switching devices.
3. One minor question : Were the quantitative measurements of the magnetization (in Fig. 2a, Supplementary Fig 4 & 6) done on films rather than on micron devices used for magnetoresistance measurements?

Once these issues are resolved or discussed in a revised manuscript, I recommend it for publication in Nature Communications.

Reviewer #2 (Remarks to the Author):

This paper demonstrates nonvolatile electric-field manipulation of magnetization in magnetic tunnel junctions deposited on a ferroelectric substrate without the help of external magnetic fields. While there are tons of papers on the electric-manipulation of magnetization with the use of ferroelectric epitaxial strain, the results are novel and thus this paper has a potential for publication in Nature Communications. However, there exist several points to be clarified and revised before considering the manuscript further for publication. In particular, the authors should describe figures and the experimental details more carefully.

1. What are the time scales of the pulsed electric field and the concomitant magnetization reversal? From the viewpoint of application, it is important to describe and discuss the time scale of the electric-field manipulation of magnetization in this system as compared with the other systems (for instance, with the help of external magnetic fields).
2. Can you estimate and discuss the potential of this system in terms of the reduction of the power consumption as compared with the similar systems?
3. Figures should be more informative. For Figs. 1c and 1d, the direction of the external magnetic field should be described. For Fig. 1f, the correspondence between the data and the double y-axes should be indicated by arrows. The illustration of magnetization in Fig. 2a should be similar to that

in Figs. 1c and 1d (red for free layer and black for pinned layer).

4. Fig. 3c contains two red arrows indicating the possible direction of ferroelectric polarization. But the directions (along [010] and [001]?) are not along the [111] equivalents, which are expected to be the polarization directions. The directions for each arrow should be appended.

5. For Figs. 2a and 2b, how did the authors estimate the magnetization in the unit of [emu/cm²] (magnetization divided by the volume of which layers)? Is the saturation value consistent with the ideal value for the magnetic layers?

Reviewer #3 (Remarks to the Author):

In their manuscript "Giant nonvolatile manipulation of magnetoresistance in magnetic tunnel junctions solely by electric fields via magneto-electric coupling" Chen and coworkers investigate the magneto-elastic manipulation of the free layer in a micrometer-sized magnetic tunnel junction. To this end, they use the strain induced on the magnetic tunnel junction by applying electric fields to the piezoelectric substrate. Most importantly, they show that manipulation of the magnetization direction of the free layer is possible even without an applied external magnetic field. For the analysis the authors compare strain and magnetization measurements as a function of the applied external electric field and find good agreement with a model based on magnetic anisotropy caused by magneto-elastic coupling. The manuscript is written in a concise way and the results warrant a publication in Nature Communications. The topic will be interesting for people working in the field of magnetoelectric control of spintronic devices. However, I have several objections to the presented work, which the authors should address in a revised version. In more detail, the authors should amend the following points in their manuscript:

1) The authors seem to be not aware of several experimental publications dating back nearly a decade that already achieved manipulation of magnetization direction by strain mediated via a piezoelectric substrate. For example: Bihler et al., Phys. Rev. B 78 , 045203 (2008, doi: 10.1103/PhysRevB.78.045203), Weiler et al., New J. Phys. 11 , 013021 (2009, doi: 10.1088/1367-2630/11/1/013021), Geprägs et al., Appl. Phys. Lett. 96 , 142509 (2010, doi: 10.1063/1.3377923), Wu et al. J. Appl. Phys. 109, 07D732 (2011, doi: 10.1063/1.3563040), Brandlmeier et al. J. of Appl. Phys. 110, 043913 (2011, doi: 10.1063/1.3624663), Wang et al., J. Appl. Phys. 114, 144902 (2013 doi: 10.1063/1.4824542). In these publications, the magneto-elastic coupling induced uniaxial anisotropy has been used to manipulate the magnetization direction of a ferromagnetic layer attached to a piezoelectric substrate. The idea to manipulate the magnetization direction without an external magnetic field has also been discussed in great length in these publications. Moreover, XRD and strain measurements have also been used in these publications to model the magneto-elastic induced magnetic anisotropy. Additionally, these experiments showed that it is possible to reversibly orient magnetization by over 70° in a nonvolatile fashion. Thus, the idea presented in this manuscript is based on already established experimental procedures. I thus suggest that the authors properly cite these pioneering works and put their results into perspective with respect to these publications in a revised version of the manuscript. Most importantly, they should highlight why the work presented in this manuscript goes beyond the already quite elaborate studies present in literature.

2) I strongly disagree with the claim of the authors that this work: "opens a promising avenue for nonvolatile electrical manipulation of MTJs". As detailed above, the idea to manipulate the magnetization direction via strain of piezoelectric substrate is not new and previous experiments showed that this fundamental principle worked. However, several issues remain that make an application for highly integrated memory or logic elements currently rather challenging. For example, the magnetization reversal in micrometer-sized junctions occurs via the nucleation and movement of magnetic domains. By reducing the size of the junction, pinning effects can lead to a large spread in necessary energies and thus may render a reliable control of all elements impossible. In addition, highly integrated devices need ways to address single cells individually, which here requires individual ferro-/piezoelectric layers underneath each MTJs. However, this has

proofed to be a tough challenge as such layers suffer from degradation, can be quite lossy, and can age very fast. Also the usage of a separate control line for addressing the magnetic cells individually increases the complexity of the integrated structure and thus the lower power consumption might be compensated by the increase in manufacturing costs. I don't see that this work solves any of these challenges. The authors should address this concern in their revised version and either provide clear evidence why the work presented here is a crucial step into this direction, or revoke/rephrase the claim accordingly.

3) The authors have nicely cited the theoretical work by Pertsev and Kohlstedt (Ref. [29]), as outlined in this work, the magnetization direction reorientation can be modelled by a free energy approach. The authors should compare the results obtained from this model with obtained electrically-controlled reorientation angles from their evaluation of the change in the TMR in their MTJs. From such an analysis one can draw conclusions, if the assumption of a single ferromagnetic domain remains valid in the micrometer-sized MTJ.

4) From the XRD measurements (Fig. 3 e) it is obvious that for the asymmetric electric field loop one obtains a different ferroelectric domain and thus strain state at zero applied electrical field. The authors should give some information on the reproducibility of this domain state as this is crucial for a reliable device operation. Moreover, in panel f of Fig. 1 for negative poling pulses it seems that a slow relaxation process over several seconds takes place before the MR settles to its final value. The authors should include these observations and discuss possible origins of these effects.

5) The authors use electric field pulses for the manipulation of the magnetization direction of the free layer. However, the applied time of these pulses is never mentioned throughout the manuscript. For future applications, it would be interesting to know how fast this switching is possible and if operation in the GHz frequency is possible. The authors should discuss this point and include, if possible, experimental evidence for short time electrical pulses inducing a magnetization reorientation.

6) Changing the electrical polarization of a ferro-/piezoelectric material requires the flow of a charge current upon the application of the electric field. As the authors claim that this approach should yield low power consumption devices, I suggest that the authors add a calculation of the power consumption for electrical switching and show that this is indeed a low power device.

Response to the reviewers' comments

We thank all the reviewers for the constructive and valuable comments on our original manuscript, which helped us greatly to improve our manuscript. In response, we have revised the manuscript thoroughly, and believe that these changes have improved our paper substantially. Below, please find our point-by-point responses to reviewers' comments as delineated in blue.

Reviewer #1 (Remarks to the Author):

The authors study the strain induced, reversible, nonvolatile electric field control of magnetization in a magnetic tunnel junction on a ferroelectric substrate. Most importantly they show a significant change in the magnetoresistance at zero field solely by application of voltage. I think these are interesting and significant experimental findings in the very technologically relevant field of low power voltage controlled spin memory devices. Therefore I recommend publication in Nature Communications after some necessary revisions are made. The issues and comments are the following:

Response: We appreciate the reviewer's comment on our paper as "Most importantly they show a significant change in the magnetoresistance at zero field solely by application of voltage. I think these are interesting and significant experimental findings in the very technologically relevant field of low power voltage controlled spin memory devices." This comment is very positive and encouraging. We have revised the original manuscript according to the reviewer's comments.

1. One problem with ferroelectric devices that the authors do not discuss is the fatigue resulting from strain mediated operations, which leads to device failure after some operations. Has the durability of these MTJ on ferroelectric devices been tested experimentally? In Fig 2b, ~50 cycles are shown, which is not enough. The durability issue is a weakness compared to "conventional" MTJ devices (e.g. Nat. Mater. 11, 64; Nat. Mater. 11, 39) which do not involve strain induced lattice distortions.

Response: For the MTJ device, as shown in Figs. 1c, 1e, 1f, we needed to measure the MR curve under electric fields, dependence of MR on electric field and MR switching by electric-field pulses. One device experienced about several hundred operations to obtain these results. In our devices, the mechanism of electric-field-controlled MR in MTJ is strain-mediated magnetoelectric coupling, so the durability of MTJ devices is good if the nonvolatile strain has an excellent endurance. We performed strain measurements under 8 kV/cm and -1.6 kV/cm electric-field pulses for more than 20000 cycles as shown in Fig. R1a. It can be seen that the nonvolatile strain is stable and reversible, suggesting that the MTJ devices should have a good endurance. In addition, the TEM image shown in Fig. S1 was obtained on a MTJ device after applying electric fields. It can be found that the MgO is crystalized and its interface is clear, so the strain did not induce lattice distortion to destroy the MTJ devices. We have added this results as Supplementary Figure 6e and Figure 6f in the Supplementary Information of the revised manuscript.

Figure R1 | **a**, The reversible and stable nonvolatile strains stitched by 8 kV/cm and -1.6 kV/cm electric-field pulses for more than 20000 cycles. **b**, The strain distribution at $\pm 0 \text{ kV/cm}$ in **a**.

2. What is the speed of the electric field manipulation of magnetoresistance in this device? In other words, how long a voltage pulse is required for this switching to occur? This is important in determining if these devices can really be utilized for low power switching, since the energy required in MTJ operations is $\text{current} \times \text{voltage} \times \text{time}$. The timescale is Fig. 1f (\sim seconds) suggests that this operation might be too slow for practical low power switching devices.

Response: The width of voltage pulse depends on the time that is required to switch the magnetization of the free layer. This time mainly includes three parts: the average magnetization switching time t_{FM} of the free layer in MTJ, the ferroelectric polarization switching time t_{FE} to produce nonvolatile strain, and the time t_{strain} for strain transfer from the ferroelectric layer to the free layer.

1), To estimate t_{FM} , we performed a macro-spin model simulation based on the conventional Landau-Lifshitz-Gilbert (LLG) equation to investigate the dynamics of the magnetization. Figure R2 presents the simulation results of magnetization rotation under electric fields. The -0 kV/cm state is obtained by applying an electric field of -1.6 kV/cm . It includes two steps, i.e., applying -1.6 kV/cm and removing it. It can be seen from Fig. R2 that the magnetization takes about 5 ns to rotate to the y direction and stabilizes after removing the electric field. The total time is less than 10 ns. Similarly, the $+0 \text{ kV/cm}$ state is obtained by applying and then removing an electric field of 8 kV/cm . From -0 kV/cm to 8 kV/cm , because the magnetic easy axes of both them are along the y direction and the magnetization almost does not change, the time for magnetization rotation is neglected as shown in Fig. R2 and the width of voltage pulse is determined by the time for ferroelectric domain switching from the in-plane to the out-of-plane. When removing electric field to $+0 \text{ kV/cm}$, the magnetization takes less than 10 ns to rotate back to the x direction. Thus the time for magnetization rotation is less than 10 ns for both -0 kV/cm state and $+0 \text{ kV/cm}$.

Figure R2 | Magnetization rotation dynamic behavior driven by an electric-field pulse. The time scale indicates the real time of magnetization evolution solved from the LLG equation.

2), The t_{FE} is usually below 10 ns and the polarization normally switches much faster than the magnetization. (Nat. Commun. 2, 553 (2011), Microelectron. Eng. 80, 296–304 (2005))

3), The t_{strain} can be estimated by $t_{strain} = \frac{d}{v}$ (J. Appl. Phys. 53, 2759 (1982)), where d is the distance between ferroelectric and the free layer, and v is the velocity of sound in the buffer layer. It is well known that the speed of sound varies from substance to substance, but generally it travels very fast in solids. We assume $v = 3000$ m/s (Nat. Commun. 2, 553 (2011)). So $t_{strain} \approx 0.009$ ns for $d = 27$ nm.

In summary, the width of voltage pulse is less than 20 ns for our devices.

For the energy dissipation in our devices, the power consumption per unit area can be estimated by CV^2/A (Nano Lett. 17, 3478-3484 (2017), Appl. Phys. Lett. 99, 063108 (2011), J. Appl. Phys. 112, 023914 (2012)), where C , V and A denote the capacitance of the piezoelectric layer, the applied voltage and the area of the device, respectively. The capacitance can be written as $C = \epsilon_r \epsilon_0 A/d$ (Nano Lett. 17, 3478-3484 (2017)), assuming a parallel plate capacitor (A is the area of the electrode, d is the thickness of the piezoelectric layer, ϵ_r is the relative dielectric constant of the piezoelectric and ϵ_0 is the vacuum dielectric constant). Thus the power consumption per unit area can be expressed as $\epsilon_r \epsilon_0 V^2/d$. In our devices, the operation electric fields for ± 0 kV/cm states in Fig. 1 are 8 kV/cm and -1.6 kV/cm, so the operation voltages are 400 V and -80 V, respectively, considering the 500 μm thickness of PMN-PT substrate. The relative dielectric constant ϵ_r of PMN-PT is about 3000 (Appl. Phys. Lett. 99, 182903 (2011)). So the power consumptions per unit area are about 0.85 mJ cm⁻² and 0.034 mJ cm⁻² for ± 0 kV cm⁻¹ states, respectively.

The state-of-the-art spin-transfer-torque magnetic tunnel junctions require about a 0.7 V voltage pulse of 500 ps (Appl. Phys. Lett 97, 242510 (2010)) or 120 ps (Appl. Phys. Lett 98, 102509 (2011)) in duration through a 60-70 nm×180 nm device, producing an energy dissipation per unit area of 3-4 mJ/cm². So the power consumption per unit area in our devices is remarkably smaller than that of the state-of-the-art spin-transfer-torque devices.

Furthermore, some theoretical work on multiferroics heterostructures using ferroelectric films has already demonstrated that fast magnetization switching can be driven by electric fields with ultralow power dissipation (Nat. Commun. 2, 553 (2011), Appl. Phys. Lett. 103, 173110 (2013), Appl. Phys. Lett. 99, 063108 (2011)). For example, energy dissipation per unit area can be reduced to 4 μJ/cm² with high-speed operation below 10 ns (Nat. Commun. 2, 553 (2011)). High quality PMN-PT epitaxial thin films on Si wafers with giant piezoelectricity have been reported (Science 334, 958-961 (2011).), which makes our work more attractive for future applications.

Our present work focus on showing that integrating spintronics and multiferroics is a reachable way to achieve nonvolatile electrical manipulation of MTJs solely by electric fields without the assistance of a magnetic field and also the potential to reduce energy consumption.

We have added a discussion on the width of voltage pulse and a calculation of the power consumption for electrical switching as Supplementary Note 3 and Note 4, respectively, in the Supplementary Information of the revised manuscript.

3. One minor question: Were the quantitative measurements of the magnetization (in Fig. 2a, Supplementary Fig 4 & 6) done on films rather than on micron devices used for magnetoresistance measurements?

Response: Yes. The magnetization was measured on films using SQUID. For micron devices, it cannot be measured using SQUID because of its small size and one useful technology for measurement is magneto-optic Kerr effect (MOKE). In our MTJ devices, the ferromagnetic layers are two 2.6 nm thick CoFeB layers and the capping layer is 5 nm Ta and 7 nm Ru. The capping layers reduce the MOKE signal of the thin magnetic layers so that it is hard to detect. Additionally, the MOKE signal is in arbitrary units so it cannot measure the variation of magnetization with electric fields as shown in Fig. 2b and Supplementary Figs. 5b, 8b and 8c. In our previous work (Sci. Rep. 4, 3727 (2014)), we investigated electric-field control of magnetism in PMN-PT/CoFeB(20 nm)/Ta(5 nm) multiferroic heterostructures using both SQUID and MOKE, and the results are similar. Thus we measured magnetization using films and the results can reflect the magnetization response to electric fields in the devices.

Once these issues are resolved or discussed in a revised manuscript, I recommend it for publication in Nature Communications.

Response: We thank the reviewer for recommending publication after addressing the issues and we hope that our revised manuscript answers the reviewer's concerns.

Reviewer #2 (Remarks to the Author):

This paper demonstrates nonvolatile electric-field manipulation of magnetization in magnetic tunnel junctions deposited on a ferroelectric substrate without the help of external magnetic fields. While there are tons of papers on the electric-manipulation of magnetization with the use of ferroelectric epitaxial strain, the results are novel and thus this paper has a potential for publication in Nature Communications. However, there exist several points to be clarified and revised before considering the manuscript further for publication. In particular, the authors should describe figures and the experimental details more carefully.

Response: We appreciate the reviewer's encouraging comment on our paper as "While there are tons of papers on the electric-manipulation of magnetization with the use of ferroelectric epitaxial strain, the results are novel and thus this paper has a potential for publication in Nature Communications." We have revised the original manuscript according to the reviewer's comments.

1. What are the time scales of the pulsed electric field and the concomitant magnetization reversal? From the viewpoint of application, it is important to describe and discuss the time scale of the electric-field manipulation of magnetization in this system as compared with the other systems (for instance, with the help of external magnetic fields).

Response: The time scales of the pulsed electric fields depend on the time that is required to switch the magnetization of the free layer. This time mainly includes three parts: the average magnetization switching time t_{FM} of the free layer in MTJ, the ferroelectric polarization switching time t_{FE} to produce nonvolatile strain, and the time t_{strain} for strain transfer from the ferroelectric layer to the free layer.

1), To estimate t_{FM} , we performed a macro-spin model simulation based on the conventional Landau-Lifshitz-Gilbert (LLG) equation to investigate the dynamics of the magnetization. Figure R3 presents the simulation results of magnetization rotation under electric fields. The -0 kV/cm state is obtained by applying an electric field of -1.6 kV/cm. It includes two steps, i.e., applying -1.6 kV/cm and removing it. It can be seen from Fig. R3 that the magnetization takes about 5 ns to rotate to the y direction and stabilizes after removing the electric field. The total time is less than 10 ns. Similarly, the $+0$ kV/cm state is obtained by applying and then removing an electric field of 8 kV/cm. From -0 kV/cm to 8 kV/cm, because the magnetic easy axes of both them are along the y direction and the magnetization almost does not change, the time for magnetization rotation is neglected as shown in Fig. R3 and the width of voltage pulse is determined by the time for ferroelectric domain switching from the in-plane to the out-of-plane. When removing electric field to $+0$ kV/cm, the magnetization takes less than 10 ns to rotate back to the x direction. Thus the time for magnetization rotation is less than 10 ns for both -0 kV/cm state and $+0$ kV/cm.

Figure R3 | Magnetization rotation dynamic behavior driven by an electric-field pulse. The time scale indicates the real time of magnetization evolution solved from the LLG equation.

2), The t_{FE} is usually below 10 ns and the polarization normally switches much faster than the magnetization. (Nat. Commun. 2, 553 (2011), Microelectron. Eng. 80, 296–304 (2005))

3), The t_{strain} can be estimated by $t_{strain} = \frac{d}{v}$ (J. Appl. Phys. 53, 2759 (1982)), where d is the distance between ferroelectric and the free layer and v is the velocity of sound in the buffer layer. It is well known that the speed of sound varies from substance to substance, but generally it travels very fast in solids. We assume $v = 3000$ m/s (Nat. Commun. 2, 553 (2011)). So $t_{strain} \approx 0.009$ ns for $d = 27$ nm.

In summary, the width of voltage pulse is less than 20 ns for our devices. In addition, the theoretical work on multiferroics heterostructures reporting that the strain-induced switching of multiferroic nanomagnets can be less than 10 ns (Nat. Commun. 2, 553 (2011), Appl. Phys. Lett. 99, 063108 (2011)) and even less than 1 ns (J. Appl. Phys. 112, 023914 (2012)). The operating speed of practical STT-MRAM devices is about 3-10 ns (AAPPS Bulletin 18, 33 (2008), J. Phys. D: Appl. Phys. 46, 074003 (2013)). So the speed of the electric-field manipulation of magnetization in multiferroic system is comparable with practical STT-MRAM devices.

We have added a discussion on the time scale of the electric-field manipulation of magnetization as Supplementary Note 3 in the Supplementary Information of the revised manuscript.

2. Can you estimate and discuss the potential of this system in terms of the reduction of the power consumption as compared with the similar systems?

Response: For the energy dissipation in our devices, the power consumption per unit area can be estimated by CV^2/A (Nano Lett. 17, 3478-3484 (2017), Appl. Phys. Lett. 99, 063108 (2011), J. Appl. Phys. 112, 023914 (2012)), where C , V and A denote the capacitance of the piezoelectric

layer, the applied voltage and the area of the device, respectively. The capacitance can be written as $C = \epsilon_r \epsilon_0 A/d$ (Nano Lett. 17, 3478-3484 (2017)), assuming a parallel plate capacitor (A is the area of the electrode, d is the thickness of the piezoelectric layer, ϵ_r is the relative dielectric constant of the piezoelectric and ϵ_0 is the vacuum dielectric constant). Thus the power consumption per unit area can be expressed as $\epsilon_r \epsilon_0 V^2/d$. In our devices, the operation electric fields for ± 0 kV/cm states in Fig. 1 are 8 kV/cm and -1.6 kV/cm, so the operation voltages are 400 V and -80 V, respectively, considering the 500 μm thickness of PMN-PT substrate. The relative dielectric constant ϵ_r of PMN-PT is about 3000 (Appl. Phys. Lett. 99, 182903 (2011)). So the power consumptions per unit area are about 0.85 mJ cm^{-2} and 0.034 mJ cm^{-2} for ± 0 kV cm^{-1} states, respectively.

The state-of-the-art spin-transfer-torque magnetic tunnel junctions require about a 0.7 V voltage pulse of 500 ps (Appl. Phys. Lett 97, 242510 (2010)) or 120 ps (Appl. Phys. Lett 98, 102509 (2011)) in duration through a 60-70 nm \times 180 nm device, producing an energy dissipation per unit area of 3-4 mJ/cm². So the power consumption per unit area in our devices is remarkably smaller than that of the state-of-the-art spin-transfer-torque devices.

Furthermore, some theoretical work on multiferroics heterostructures using ferroelectric films has already demonstrated that fast magnetization switching can be driven by electric fields with ultralow power dissipation (Nat. Commun. 2, 553 (2011), Appl. Phys. Lett. 103, 173110 (2013), Appl. Phys. Lett. 99, 063108 (2011)). For example, energy dissipation per unit area can be reduced to 4 $\mu\text{J/cm}^2$ with high-speed operation below 10 ns (Nat. Commun. 2, 553 (2011)). High quality PMN-PT epitaxial thin films on Si wafers with giant piezoelectricity have been reported (Science 334, 958-961 (2011).), which makes our work more attractive for future applications.

Our present work focus on showing that integrating spintronics and multiferroics is a reachable way to achieve nonvolatile electrical manipulation of MTJs solely by electric fields without the assistance of a magnetic field and also the potential to reduce energy consumption.

We have added a calculation of the power consumption as Supplementary Note 4 in the Supplementary Information of the revised manuscript.

3. Figures should be more informative. For Figs. 1c and 1d, the direction of the external magnetic field should be described. For Fig. 1f, the correspondence between the data and the double y-axes should be indicated by arrows. The illustration of magnetization in Fig. 2a should be similar to that in Figs. 1c and 1d (red for free layer and black for pinned layer).

Response: Thank the reviewer for very careful reading. Following the reviewer's valuable suggestion, we have revised the figures. For the illustration of magnetization in Figs. 1c and 1d, the dark arrow denotes the magnetization of the pinned layer. The red and blue arrows denote the magnetization of the free layer at $E = +0$ kV/cm and -0 kV/cm, respectively.

4. Fig. 3c contains two red arrows indicating the possible direction of ferroelectric polarization. But the directions (along [010] and [001]?) are not along the [111] equivalents, which are expected to be the polarization directions. The directions for each arrow should be appended.

Response: We are sorry for making reviewer misunderstanding. The rhombohedral crystal structure of PMN-PT (011) has eight spontaneous ferroelectric polarization directions along the $\langle 111 \rangle$ directions (the body diagonals of the pseudocubic unit cell). We revised Fig. 3c to make it clear and appended the directions for each polarization.

5. For Figs. 2a and 2b, how did the authors estimate the magnetization in the unit of $[\text{emu}/\text{cm}^2]$ (magnetization divided by the volume of which layers)? Is the saturation value consistent with the ideal value for the magnetic layers?

Response: For magnetic measurements by SQUID, the sample structure is Ta(5)/Cu(10)/Ta(5)/IrMn(8)/CoFeB(4)/MgO(2.2)/CoFeB(4)/Ta(5)/Cu(5)/Ta(5) (the unit is nanometer) which has two ferromagnetic CoFeB layer and the sample size is $5 \text{ mm} \times 5 \text{ mm}$. The magnetization is estimated through dividing measured magnetic moment by the volume of two ferromagnetic layer ($5 \text{ mm} \times 5 \text{ mm} \times 8 \text{ nm}$). The saturation magnetization is about $1200 \text{ emu}/\text{cm}^3$. This value is consistent with that of single CoFeB layer shown in Supplementary Figs. 7 and 8, and is similar to that of previous papers (e.g., Sci. Rep. 4, 3727 (2014), Phy. Rev. B 83, 212404 (2011), Adv. Mater. 26, 4320 (2014), Phy. Rev. B 95, 104435 (2017)).

Reviewer #3 (Remarks to the Author):

In their manuscript “Giant nonvolatile manipulation of magnetoresistance in magnetic tunnel junctions solely by electric fields via magneto-electric coupling” Chen and coworkers investigate the magneto-elastic manipulation of the free layer in a micrometer-sized magnetic tunnel junction. To this end, they use the strain induced on the magnetic tunnel junction by applying electric fields to the piezoelectric substrate. Most importantly, they show that manipulation of the magnetization direction of the free layer is possible even without an applied external magnetic field. For the analysis the authors compare strain and magnetization measurements as a function of the applied external electric field and find good agreement with a model based on magnetic anisotropy caused by magneto-elastic coupling. The manuscript is written in a concise way and the results warrant a publication in Nature Communications. The topic will be interesting for people working in the field of magnetoelectric control of spintronic devices. However, I have several objections to the presented work, which the authors should address in a revised version. In more detail, the authors should amend the following points in their manuscript:

Response: We appreciate the reviewer’s encouraging comment on our paper as “The manuscript is written in a concise way and the results warrant a publication in Nature Communications. The topic will be interesting for people working in the field of magnetoelectric control of spintronic devices.” We have revised the original manuscript according to the reviewer’s comments.

1) The authors seem to be not aware of several experimental publications dating back nearly a decade that already achieved manipulation of magnetization direction by strain mediated via a piezoelectric substrate. For example: Bihler et al., Phys. Rev. B 78 , 045203 (2008, doi: 10.1103/PhysRevB.78.045203), Weiler et al., New J. Phys. 11 , 013021 (2009, doi: 10.1088/1367-2630/11/1/013021), Geprägs et al., Appl. Phys. Lett. 96 , 142509 (2010, doi: 10.1063/1.3377923), Wu et al. J. Appl. Phys. 109, 07D732 (2011, doi: 10.1063/1.3563040), Brandlmeier et al. J. of Appl. Phys. 110, 043913 (2011, doi: 10.1063/1.3624663), Wang et al., J. Appl. Phys. 114, 144902 (2013 doi: 10.1063/1.4824542). In these publications, the magneto-elastic coupling induced uniaxial anisotropy has been used to manipulate the magnetization direction of a ferromagnetic layer attached to a piezoelectric substrate. The idea to manipulate the magnetization direction without an external magnetic field has also been discussed in great length in these publications. Moreover, XRD and strain measurements have also been used in these publications to model the magneto-elastic induced magnetic anisotropy. Additionally, these experiments showed that it is possible to reversibly orient magnetization by over 70° in a nonvolatile fashion. Thus, the idea presented in this manuscript is based on already established experimental procedures. I thus suggest that the authors properly cite these pioneering works and put their results into perspective with respect to these publications in a revised version of the manuscript. Most importantly, they should highlight why the work presented in this manuscript goes beyond the already quite elaborate studies present in literature.

Response: We agree with the reviewer that there is a lot of work on electric-field control of magnetism in ferromagnetic/ferroelectric multiferroic heterostructures in the past decade. We are sorry that we did not cite them in the original manuscript, which did not mean that they are not important, because we were not able to put all of them in one paper. So we only cited some representative reviews (such as Refs. 14, 18-22, 30-34) in order to give the readers a

comprehensive view of this research direction. According to the reviewer's suggestion, we have added these references (Refs. 35-40) in the revised manuscript.

As mentioned in the introduction part of our paper, "Nonvolatile electrically controlled magnetism was demonstrated in multiferroic heterostructures, however, it has not been used in spintronic devices." The previous reports in the literature mainly focused on tuning magnetism by electric fields in ferromagnetic/ferroelectric multiferroic heterostructures with ferromagnetic thin films, but it was rare to extend the work to spintronic devices such as magnetic tunnel junctions. One of the motivations for multiferroics research is to find a low-energy solution to spintronics applications while the experiment work is rare. In our work, we focused on integrating spintronics with multiferroics by depositing magnetic tunnel junctions on ferroelectric substrate. Utilizing strain-mediated magnetoelectric coupling, we experimentally demonstrated electrical control of magnetoresistance in magnetic tunnel junctions at zero magnetic field. As far as we know, this is the first report on a room temperature, giant and nonvolatile manipulation of magnetoresistance in MTJs controlled solely by electric fields without the assistance of a magnetic field, which makes our work quite unique and is also an important step for applications of multiferroic heterostructures.

2) I strongly disagree with the claim of the authors that this work: "opens a promising avenue for nonvolatile electrical manipulation of MTJs". As detailed above, the idea to manipulate the magnetization direction via strain of piezoelectric substrate is not new and previous experiments showed that this fundamental principle worked. However, several issues remain that make an application for highly integrated memory or logic elements currently rather challenging. For example, the magnetization reversal in micrometer-sized junctions occurs via the nucleation and movement of magnetic domains. By reducing the size of the junction, pinning effects can lead to a large spread in necessary energies and thus may render a reliable control of all elements impossible. In addition, highly integrated devices need ways to address single cells individually, which here requires individual ferro-/piezoelectric layers underneath each MTJs. However, this has proved to be a tough challenge as such layers suffer from degradation, can be quite lossy, and can age very fast. Also the usage of a separate control line for addressing the magnetic cells individually increases the complexity of the integrated structure and thus the lower power consumption might be compensated by the increase in manufacturing costs. I don't see that this work solves any of these challenges. The authors should address this concern in their revised version and either provide clear evidence why the work presented here is a crucial step into this direction, or revoke/rephrase the claim accordingly.

Response: Indeed, the manipulation of magnetization by strain-mediated magnetoelectric coupling has been reported in ferroelectric/ferromagnetic multiferroic heterostructures. However, these work mostly investigated ferromagnetic films instead of devices such as MTJs. Though nonvolatile electrical control of MTJs was studied theoretically, the corresponding experiment remained elusive in the past years. Our work demonstrates nonvolatile manipulation of magnetoresistance in MTJs controlled solely by electric fields without the assistance of a magnetic field, which shows the previous theory can be experimentally realized. It should be mentioned that the statement of our work "opens a promising avenue for nonvolatile electrical

manipulation of MTJs” is referred to the comparison with the previous reports on controlling MTJs with electric fields via other ways (Nat. Mater. 7, 425 (2008), Nat. Commun. 2, 553 (2011), Nat. Mater. 11, 64 (2012), Nat. Mater. 11, 39 (2012), Nat. Mater. 6, 296 (2007), Science 327, 1106 (2010), Nat. Mater. 11, 289 (2012)). The referee misunderstood the meaning of this statement.

The issues mentioned by the reviewer exist in current spintronics, such as the micrometer-sized junctions suffering from domain nucleation and movement, and the pinning effect in nanometer junctions (Nat. Mater. 6, 813-823 (2007), Appl. Phys. Lett. 99, 042501 (2011)). Finding these solutions is beyond the investigation in our present work. Controlling MTJs with electric fields is also a very important issue, and our present work focused on showing that integrating spintronics and multiferroics is a reachable way to achieve nonvolatile electrical manipulation of MTJs solely by electric fields without the assistance of a magnetic field and also the potential to reduce energy consumption.

As the reviewer said, for future highly integrated devices, individual ferro-/piezoelectric layers underneath each MTJs are required. According to the work by S. H. Baek et al. (Nature Mater. 9, 309 (2010)), reducing the BiFeO₃ island to 100 nm × 100 nm effectively enhances the stability of the ferroelectric domain. So the individual ferro-/piezoelectric nanoisland can be utilized in MTJ/ferroelectric structure.

For MTJs/ferroelectric devices, the three-terminal design of this scheme separates the write and read cells. Figure R4 shows the schematics of a 1-transistor, 1-MTJ memory cell for conventional MRAM, STT-MRAM, SOT-MRAM and MTJs/ferroelectric. The conventional MRAM uses the magnetic field generated by current to write information and it has two current lines. The STT-MRAM is two-terminal device while SOT-MRAM and MTJs/ferroelectric are three-terminal devices. It can be seen that the structures of SOT-MRAM and MTJs/ferroelectric are similar. Due to the absence of extra current-carrying lines, the MTJs/ferroelectric structure has a simpler structure and is easier to manufacture than the conventional MRAMs, thus it will not increase the manufacturing cost.

Figure R4 | Schematics of a 1-transistor, 1-MTJ memory cell for **a**, conventional MRAM, **b**, STT-MRAM, **c**, SOT-MRAM and **d**, MTJs/ferroelectric. **a** is reprinted from Nature Mater. 6, 813 (2007) and **b,c** are reprinted from Mater. Today 20, 530 (2017).

In our work, we successfully fabricated MTJs on PMN-PT ferroelectric substrate and experimentally realized nonvolatile manipulation of MTJ solely by electric fields without the assistance of a magnetic field. These results show that integrating multiferroics and spintronics is

feasible. Moreover, high quality PMN-PT epitaxial thin films on Si wafers with giant piezoelectricity have been reported (Science 334, 958-961 (2011)). So for future applications, MTJ can be grown on PMN-PT film, which offers a prospective approach to integrate MTJs/FE structure with current silicon-based electronics. Our present work based on PMN-PT substrate gives a prelude to this attractive future application.

According to the reviewer's comment, we revised "Our work opens a promising avenue for nonvolatile electrical manipulation of MTJs" to "Our work experimentally realizes nonvolatile electrical manipulation of MTJs by integrating multiferroics and spintronics" to make it more specific.

3) The authors have nicely cited the theoretical work by Pertsev and Kohlstedt (Ref. [29]), as outlined in this work, the magnetization direction reorientation can be modelled by a free energy approach. The authors should compare the results obtained from this model with obtained electrically-controlled reorientation angles from their evaluation of the change in the TMR in their MTJs. From such an analysis one can draw conclusions, if the assumption of a single ferromagnetic domain remains valid in the micrometer-sized MTJ.

Response: In Ref. 29, Pertsev and Kohlstedt used free energy model to study the magnetic tunnel junction on a ferroelectric substrate. In our system, the free energy density can be expressed as

$$E = -K_0 \cos^2 \alpha - K_{\text{strain}} \cos^2 (\alpha - 90^\circ) = -K_{\text{strain}} + (K_{\text{strain}} - K_0) \cos^2 \alpha = -\frac{M_S H_0}{2} + \frac{M_S}{2} (H_{\text{eff, S}} - H_0) \cos^2 \alpha$$

Here, M_S is the saturation magnetization of the free layer with 1200 emu/cm^3 , α is the angle of magnetic moment with respect to the [100] direction of PMN-PT, H_0 (about 90 Oe) is the magnetic anisotropy field resulting from the initial randomness of ferroelectric domains in PMN-PT (Sci. Rep. 4, 3727 (2014)), and $H_{\text{eff, S}}$ is the strain-induced effective anisotropy field as shown in Supplementary Fig. 7c.

Figure R5 | Free energy density of the free layer as a function of the magnetization orientation for $E = \pm 0$ kV/cm.

Fig. R5 shows the free energy profile at $E = \pm 0$ kV/cm. For $E = +0$ kV/cm, the energy minimizes at $\alpha = 0^\circ$ so the magnetization prefers to point to the x direction. For $E = -0$ kV/cm, the energy minimizes at $\alpha = 90^\circ$ so the magnetization prefers to point to the y direction. In our micro-sized MTJ, this free energy model can qualitatively simulate the change of magnetic easy axis under electric fields, but it is too simple to simulate the magnetization orientation.

Moreover, we performed micromagnetic simulations using the object-oriented micromagnetic framework (OOMMF) software to study the domain evolution of the free layer under electric fields. Similar to the device structure, a circular shaped CoFeB disk of $10 \mu\text{m}$ in diameter was built and discretized in the computational cells of $10 \text{ nm} \times 10 \text{ nm}$. The saturation magnetization, M_S , is 1200 emu/cm^3 ; and the uniform exchange constant, A , is $2.8 \times 10^{-11} \text{ J/m}$ (ACS Appl. Mater. Inter. 9, 2642 (2017)). To simulate the domain structures of the free layer of the MTJ under electric fields, a single domain along the $[100]$ direction of PMN-PT (the pinning direction of the MTJs) was set as the initial state. Figure R6 shows the domain structures at $E = \pm 0$ kV/cm. For $E = +0$ kV/cm, the magnetization mainly points to the x direction while the magnetization rotates to the y direction for $E = -0$ kV/cm. But the magnetization around the edge is not rotated by 90° . This may be one reason that the rotation angles of the free layer at $H = 0$ Oe for $E = \pm 0$ kV/cm are not actually 90° as shown in Supplementary Table 1. Additionally, the TMR curve at $E = +0$ kV/cm (Fig. 1c) is not square and sharp suggesting the existence of multidomain in the micrometer-sized circular-shaped MTJ. This also can lead to the rotation angle of the free layer smaller than 90° .

Figure R6 | Micromagnetic simulation of ferromagnetic domain structure of the free layer at **a**, $E = +0$ kV/cm and **b**, $E = -0$ kV/cm.

So the assumption of a single ferromagnetic domain can roughly remain valid in the micrometer-sized MTJ. We have added this micromagnetic simulation as Supplementary Figure 4 in the Supplementary Information of the revised manuscript.

4) From the XRD measurements (Fig. 3 e) it is obvious that for the asymmetric electric field loop one obtains a different ferroelectric domain and thus strain state at zero applied electrical field. The authors should give some information on the reproducibility of this domain state as this is crucial for a reliable device operation. Moreover, in panel f of Fig. 1 for negative poling pulses it seems that a slow relaxation process over several seconds takes place before the MR settles to its final value. The authors should include these observations and discuss possible origins of these effects.

Response: The ferroelectric domain state determines the strain state, so we measured the endurance of the nonvolatile strain under 8 kV/cm and -1.6 kV/cm electric-field pulses to show the reproducibility of this domain state. Figure R7a shows switching of nonvolatile strain under 8 kV/cm and -1.6 kV/cm electric-field pulses. After more than 20000 cycles, the nonvolatile strain is stable and reversible. The statistics of strain for ± 0 kV/cm states is shown in Figure R7b. This reversible and stable nonvolatile strain suggests the well reproducibility of ferroelectric domain state. The results shows the nonvolatile strain is reproducible, suggesting the reproducibility of the in-plane and out-of-plane ferroelectric polarization for ± 0 kV/cm states. We have added this results as Supplementary Figure 6e and Figure 6f in the Supplementary Information of the revised manuscript.

Figure R7 | **a**, The reversible and stable nonvolatile strains stitched by 8 kV/cm and -1.6 kV/cm electric-field pulses for more than 20000 cycles. **b**, The strain distribution at ± 0 kV/cm in **a**.

The slow relaxation process in Fig. 1 may originate from the release process of the charges induced by ferroelectric polarization. This resistance relaxation is also reported in the literatures such as Adv. Mater. 26, 4320-4325 (2014), Appl. Phys. Lett. 109, 092403 (2016), Appl. Phys. Lett 98, 222509 (2011). For negative poling pulses, the ferroelectric polarization reorients to the in-plane metastable state (Appl. Phys. Lett. 98, 012504 (2011)), which may also lead to the slow relaxation process of magnetoresistance. We have added this discussion in the page 7 of the main text in the revised manuscript.

5) The authors use electric field pulses for the manipulation of the magnetization direction of the free layer. However, the applied time of these pulses is never mentioned throughout the manuscript. For future applications, it would be interesting to know how fast this switching is

possible and if operation in the GHz frequency is possible. The authors should discuss this point and include, if possible, experimental evidence for short time electrical pulses inducing a magnetization reorientation.

Response: The time scales of the pulsed electric fields depend on the time that is required to switch the magnetization of the free layer. This time mainly includes three parts: the average magnetization switching time t_{FM} of the free layer in MTJ, the ferroelectric polarization switching time t_{FE} to produce nonvolatile strain, and the time t_{strain} for strain transfer from the ferroelectric layer to the free layer.

1), To estimate t_{FM} , we performed a macro-spin model simulation based on the conventional Landau-Lifshitz-Gilbert (LLG) equation to investigate the dynamics of the magnetization. Figure R8 presents the simulation results of magnetization rotation under electric fields. The -0 kV/cm state is obtained by applying an electric field of -1.6 kV/cm. It includes two steps, i.e., applying -1.6 kV/cm and removing it. It can be seen from Fig. R8 that the magnetization takes about 5 ns to rotate to the y direction and stabilizes after removing the electric field. The total time is less than 10 ns. Similarly, the $+0$ kV/cm state is obtained by applying and then removing an electric field of 8 kV/cm. From -0 kV/cm to 8 kV/cm, because the magnetic easy axes of both them are along the y direction and the magnetization almost does not change, the time for magnetization rotation is neglected as shown in Fig. R8 and the width of voltage pulse is determined by the time for ferroelectric domain switching from the in-plane to the out-of-plane. When removing electric field to $+0$ kV/cm, the magnetization takes less than 10 ns to rotate back to the x direction. Thus the time for magnetization rotation is less than 10 ns for both -0 kV/cm state and $+0$ kV/cm.

Figure R8 | Magnetization rotation dynamic behavior driven by an electric-field pulse. The time scale indicates the real time of magnetization evolution solved from the LLG equation.

2), The t_{FE} is usually below 10 ns and the polarization normally switches much faster than the magnetization. (Nat. Commun. 2, 553 (2011), Microelectron. Eng. 80, 296–304 (2005))

3), The t_{strain} can be estimated by $t_{strain} = \frac{d}{v}$ (J. Appl. Phys. 53, 2759 (1982)), where d is the distance between ferroelectric and the free layer and v is the velocity of sound in the buffer layer. It is well known that the speed of sound varies from substance to substance, but generally it travels very fast in solids. We assume $v = 3000$ m/s (Nat. Commun. 2, 553 (2011)). So $t_{strain} \approx 0.009$ ns for $d = 27$ nm.

In summary, the width of voltage pulse is less than 20 ns for our devices. In addition, the theoretical work on multiferroics heterostructures reporting that the strain-induced switching of multiferroic nanomagnets can be less than 10 ns (Nat. Commun. 2, 553 (2011), Appl. Phys. Lett. 99, 063108 (2011)) and even less than 1 ns (J. Appl. Phys. 112, 023914 (2012)). So the future operation in the GHz frequency should be possible.

We have added a discussion on the time scale of the electric-field manipulation of magnetization as Supplementary Note 3 in the Supplementary Information of the revised manuscript.

6) Changing the electrical polarization of a ferro-/piezoelectric material requires the flow of a charge current upon the application of the electric field. As the authors claim that this approach should yield low power consumption devices, I suggest that the authors add a calculation of the power consumption for electrical switching and show that this is indeed a low power device.

Response: For the energy dissipation in our devices, the power consumption per unit area can be estimated by CV^2/A (Nano Lett. 17, 3478-3484 (2017), Appl. Phys. Lett. 99, 063108 (2011), J. Appl. Phys. 112, 023914 (2012)), where C , V and A denote the capacitance of the piezoelectric layer, the applied voltage and the area of the device, respectively. The capacitance can be written as $C = \epsilon_r \epsilon_0 A/d$ (Nano Lett. 17, 3478-3484 (2017)), assuming a parallel plate capacitor (A is the area of the electrode, d is the thickness of the piezoelectric layer, ϵ_r is the relative dielectric constant of the piezoelectric and ϵ_0 is the vacuum dielectric constant). Thus the power consumption per unit area can be expressed as $\epsilon_r \epsilon_0 V^2/d$. In our devices, the operation electric fields for ± 0 kV/cm states in Fig. 1 are 8 kV/cm and -1.6 kV/cm, so the operation voltages are 400 V and -80 V, respectively, considering the 500 μm thickness of PMN-PT substrate. The relative dielectric constant ϵ_r of PMN-PT is about 3000 (Appl. Phys. Lett. 99, 182903 (2011)). So the power consumptions per unit area are about 0.85 mJ cm^{-2} and 0.034 mJ cm^{-2} for ± 0 kV cm^{-1} states, respectively.

The state-of-the-art spin-transfer-torque magnetic tunnel junctions require about a 0.7 V voltage pulse of 500 ps (Appl. Phys. Lett 97, 242510 (2010)) or 120 ps (Appl. Phys. Lett 98, 102509 (2011)) in duration through a 60-70 nm \times 180 nm device, producing an energy dissipation per unit area of 3-4 mJ/cm². So the power consumption per unit area in our devices is remarkably smaller than that of the state-of-the-art spin-transfer-torque devices.

Furthermore, some theoretical work on multiferroics heterostructures using ferroelectric films has already demonstrated that fast magnetization switching can be driven by electric fields with ultralow power dissipation (Nat. Commun. 2, 553 (2011), Appl. Phys. Lett. 103, 173110 (2013),

Appl. Phys. Lett. 99, 063108 (2011)). For example, energy dissipation per unit area can be reduced to $4 \mu\text{J}/\text{cm}^2$ with high-speed operation below 10 ns (Nat. Commun. 2, 553 (2011)). High quality PMN-PT epitaxial thin films on Si wafers with giant piezoelectricity have been reported (Science 334, 958-961 (2011).), which makes our work more attractive for future applications

We have added a calculation of the power consumption as Supplementary Note 4 in the Supplementary Information of the revised manuscript.

A summary of main changes in the revised manuscript

According to the suggestions and comments of the reviewers, we have made changes as follows:

1. According to the comment 1 of reviewer #1 that “One problem with ferroelectric devices that the authors do not discuss is the fatigue resulting from strain mediated operations” and the comment 2 of reviewer #3 that “The authors should give some information on the reproducibility of this domain state as this is crucial for a reliable device operation”, we have performed strain measurements under 8 kV/cm and -1.6 kV/cm electric-field pulses for more than 20000 cycles and added these results as Supplementary Figure 6e and Figure 6f in the Supplementary Information of the revised manuscript.
2. According to the comment 2 of reviewer #1, the comment 1 of reviewer #2 and the comment 5 of reviewer #3, we have added a discussion on the time scale of the electric-field manipulation of magnetization as Supplementary Note 3 in the Supplementary Information of the revised manuscript.
3. According to the comment 2 of reviewer #1, the comment 2 of reviewer #2 and the comment 6 of reviewer #3, we have added a calculation of the power consumption as Supplementary Note 4 in the Supplementary Information of the revised manuscript.
4. As suggested by reviewer #2 in comment 3 and comment 4, we revised the figures. In Figs. 1c and 1d, the direction of the external magnetic field has been described. In Fig. 1f, the correspondence between the data and the double y-axes has been indicated by arrows. The illustration of magnetization in Fig. 2a was revised to be similar to that in Figs. 1c and 1d. In Figs. 3c and 3d, the directions of the ferroelectric polarization have been added to make it clear.
5. As suggested by reviewer #3 in comment 1, we have added new references (Refs. 35-40) in the main text of the revised manuscript.
6. According to the comment 2 of reviewer #3, we revised “Our work opens a promising avenue for nonvolatile electrical manipulation of MTJs” to “Our work experimentally realizes nonvolatile electrical manipulation of MTJs by integrating multiferroics and spintronics”.
7. According to the comment 3 of reviewer #3, we performed micromagnetic simulations to study the domain evolution of the free layer under electric fields and added these results as Supplementary Figure 4 in the Supplementary Information of the revised manuscript.
8. As suggested by reviewer #3 in comment 4 that “Moreover, in panel f of Fig. 1 for negative poling pulses it seems that a slow relaxation process over several seconds takes place before the MR settles to its final value. The authors should include these observations and discuss possible origins of these effects.” We have added an explanation on Page 7 of the main text of the revised manuscript.
9. To comply with the format requirements, there are three other changes. (1) “kV/cm” was changed to “kV cm⁻¹”. (2) “[01-1]” was changed to “[01 $\bar{1}$]”. (3) The title “Giant nonvolatile manipulation of magnetoresistance in magnetic tunnel junctions solely by electric fields via

magnetoelectric coupling” was changed to “Giant nonvolatile manipulation of magnetoresistance in magnetic tunnel junctions by electric fields via magnetoelectric coupling”, because the title is not more than 15 words.

REVIEWERS' COMMENTS:

Reviewer #1 (Remarks to the Author):

The authors have answered the issues raised and revised the manuscript according to the comments. Although it would be much better if the fast switching could be shown experimentally (related to comment 2), rather than giving a theoretical estimate with the LLG equation, I believe the revision is satisfactory.

Reviewer #2 (Remarks to the Author):

The authors have complied with the comments raised by three reviewers. The manuscript is ready for publication.

Reviewer #3 (Remarks to the Author):

In the revised version of their manuscript "Giant nonvolatile manipulation of magnetoresistance in magnetic tunnel junctions by electric fields via magnetoelectric coupling" the authors have nicely addressed all raised concerns of the three reviewers. The added supplementary information provides a much deeper insight into this interesting new direction for low power, voltage controlled magnetic memories. Especially, the presented additional micromagnetic simulations and LLG simulations provide a better understanding of the physical limits of these devices. The manuscript in its present state warrants a publication in Nature Communications. However, I recommend the authors make one final minor addition to their nice work before being accepted for publication:

The authors now nicely estimate the minimal switching time for their magnetic tunnel junction from LLG simulations. I really find this theoretical estimation rather intriguing. However, in their experiments the voltage pulses used are clearly longer than a couple of 10 ns (at least Figure 1 f suggest that the pulses are several seconds long). To reflect this theoretical conjecture of short switching times, I suggest that the authors add the duration of the used voltage pulses to the methods section of their manuscript and add a short sentences that this high frequency operation has to be verified in future experiments to the main text.

Response to the reviewers' comments

Reviewer #1 (Remarks to the Author):

The authors have answered the issues raised and revised the manuscript according to the comments. Although it would be much better if the fast switching could be shown experimentally (related to comment 2), rather than giving a theoretical estimate with the LLG equation, I believe the revision is satisfactory.

Response: We are delighted that our reply answered the reviewer's concerns and appreciate the reviewer's comment that "I believe the revision is satisfactory".

Reviewer #2 (Remarks to the Author):

The authors have complied with the comments raised by three reviewers. The manuscript is ready for publication.

Response: We are delighted that our reply answered the reviewer's concerns and appreciate the reviewer's comment that "The manuscript is ready for publication".

Reviewer #3 (Remarks to the Author):

In the revised version of their manuscript "Giant nonvolatile manipulation of magnetoresistance in magnetic tunnel junctions by electric fields via magnetoelectric coupling" the authors have nicely addressed all raised concerns of the three reviewers. The added supplementary information provides a much deeper insight into this interesting new direction for low power, voltage controlled magnetic memories. Especially, the presented additional micromagnetic simulations and LLG simulations provide a better understanding of the physical limits of these devices. The manuscript in its present state warrants a publication in Nature Communications. However, I recommend the authors make one final minor addition to their nice work before being accepted for publication: The authors now nicely estimate the minimal switching time for their magnetic tunnel junction from LLG simulations. I really find this theoretical estimation rather intriguing. However, in their experiments the voltage pulses used are clearly longer than a couple of 10 ns (at least Figure 1 f suggest that the pulses are several seconds long). To reflect this theoretical conjecture of short switching times, I suggest that the authors add the duration of the used voltage pulses to the methods section of their manuscript and add a short sentences that this high frequency operation has to be verified in future experiments to the main text.

Response: We are delighted that our reply answered the reviewer's concerns and appreciate the reviewer's comment that "The manuscript in its present state warrants a publication in Nature Communications". We thank the reviewer's suggestion and have added a discussion in the methods section.